# Home-EEG assessment of possible compensatory mechanisms for sleep disruption in highly irregular shift workers – The ANCHOR study

**Lara J. Mentink**[1,2,3‡]*, **Jana Thomas**[1,2,3‡], **René J. F. Melis**[1,2,4], **Marcel G. M. Olde Rikkert**[1,2,3], **Sebastiaan Overeem**[5,6], **Jurgen A. H. R. Claassen**[1,2,3]

1 Department of Geriatric Medicine, Radboud University Medical Center, Nijmegen, The Netherlands, 2 Radboud Alzheimer Centre, Radboud University Medical Center, Nijmegen, The Netherlands, 3 Donders Institute for Brain, Cognition and Behaviour, Radboud University, Nijmegen, The Netherlands, 4 Radboud Institute for Health Sciences, Radboud University Medical Center, Nijmegen, The Netherlands, 5 Sleep Medicine Center Kempenhaeghe, Heeze, The Netherlands, 6 Biomedical Diagnostics Group, Department of Electrical Engineering, Eindhoven University of Technology, Eindhoven, The Netherlands

‡ Dual first authorship: The two first authors have contributed equally to the manuscript.
* Lara.Mentink@radboudumc.nl

**Data Availability Statement:** The data has been published in the DANS EASY repository, which is the Netherlands Institute for permanent access to

## Abstract

### Study objectives

While poor sleep quality has been related to increased risk of Alzheimer's disease, long-time shift workers (maritime pilots) did not manifest evidence of early Alzheimer's disease in a recent study. We explored two hypotheses of possible compensatory mechanisms for sleep disruption: Increased efficiency in generating deep sleep during workweeks (model 1) and rebound sleep during rest weeks (model 2).

### Methods

We used data from ten male maritime pilots (mean age: 51.6±2.4 years) with a history of approximately 18 years of irregular shift work. Subjective sleep quality was assessed with the Pittsburgh Sleep Quality Index (PSQI). A single lead EEG-device was used to investigate sleep in the home/work environment, quantifying total sleep time (TST), deep sleep time (DST), and deep sleep time percentage (DST%). Using multilevel models, we studied the sleep architecture of maritime pilots over time, at the transition of a workweek to a rest week.

### Results

Maritime pilots reported worse sleep quality in workweeks compared to rest weeks (PSQI = 8.2±2.2 vs. 3.9±2.0; p<0.001). Model 1 showed a trend towards an increase in DST% of 0.6% per day during the workweek (p = 0.08). Model 2 did not display an increase in DST% in the rest week (p = 0.87).

digital research resources. The data are published under the following DOI: https://doi.org/10.17026/dans-x6b-km62.

**Funding:** This study was funded by the ISAO grant (Internationale Stichting Alzheimer Onderzoek, grant number: 15040), received by JC, and internal funding by the Radboud University Medical Center, received by JC & MOR. There was no additional external funding received for this study. The funders had no role in study design, data collection and analysis, decision to publish, or preparation of the manuscript.

**Competing interests:** The authors declare that Philips (Royal Philips, Amsterdam, The Netherlands) kindly lent us the home-EEG devices that were used in this study. This does not alter our adherence to PLOS ONE policies on sharing data and materials. Philips had no role in study design, data collection and analysis, decision to publish, or preparation of the manuscript.

## Conclusions

Our findings indicated that increased efficiency in generating deep sleep during workweeks is a more likely compensatory mechanism for sleep disruption in the maritime pilot cohort than rebound sleep during rest weeks. Compensatory mechanisms for poor sleep quality might mitigate sleep disruption-related risk of developing Alzheimer's disease. These results should be used as a starting point for future studies including larger, more diverse populations of shift workers.

## Introduction

Sleep disruption has been associated with higher risks of developing Alzheimer's disease (AD) [1–5]. In recent studies, individuals with sleep problems carried a 1.7 (95% CI 1.5 to 1.9) higher relative dementia risk compared to normal sleepers [6], suggesting that 15% of current AD diagnoses might be attributable to sleep problems [7]. One of the hallmarks of Alzheimer's pathology is the accumulation of amyloid-β, which is a potential mechanistic link between AD and sleep [8–13]. During wakefulness, amyloid-β builds up in the brain which is hypothesized to be counteracted during deep sleep in two ways; through improved clearance of accumulated toxins (such as amyloid-β), driven by the glymphatic system [8–11, 14] or due to an overall reduced level of synaptic activity in the brain, leading to a decrease in production of waste products (such as amyloid-β) [13, 15, 16]. The reduced level of brain activity during deep sleep also leads to less blood flow and more cerebrospinal fluid (CSF) flow, which additionally intensifies clearance of accumulated waste products [12, 17]. These hypotheses indicate how, through accumulation of amyloid-β, poor sleep could be a causal risk factor for AD.

Indeed, studies reported increased amyloid-β concentration in CSF [18] and an acute increase of amyloid-β assessed with PET and MRI [19] after one night of sleep deprivation compared to unrestricted sleep. Selective disruption of deep sleep without affecting other sleep stages led to a comparable increase in amyloid-β concentration in CSF [20]. Previous studies mostly investigated acute effects of sleep deprivation, whereas effects of long-term exposure to poor sleep has not been studied extensively before. The SCHIP study (Sleep-Cognition-Hypothesis In maritime Pilots) conducted by our group in 2016–2019, hypothesized that long-term exposure to sleep disruption leads to an increased AD-risk [21]. The maritime pilots included in the SCHIP study follow work schedules, characterized by one week with irregular working hours, resulting in a combination of sleep restriction, fragmentation and deprivation, followed by a rest week with unrestricted sleep. We found that maritime pilots, with an average of 18 years of irregular and unpredictable work shifts (night & day) did not manifest any AD-evidence, such as cognitive deficits or brain amyloid-β accumulation [22].

In a separate study, we found that retired maritime pilots, who had worked irregular shifts for approximately 26 years did not show any signs of early dementia or MCI [23]. Neither did the long-term exposure to irregular shift work result in circadian rhythm disruption, mood complaints or decreased quality of life after employment [23]. Results of these two studies are in contrast with earlier studies claiming that sleep loss leads to higher brain amyloid-β concentrations and cognitive decline.

In the present study, the ANCHOR study (Assessing Nightly Components Highly Operative to Recovery), we investigated potential causes for the absence of amyloid-β accumulation and cognitive dysfunction after long-term exposure to sleep disruption in this specific cohort.

By using a novel, wearable home-EEG device, we studied sleep architecture of maritime pilots during and immediately after a workweek. The findings of previous studies led to two hypotheses; first, we hypothesize that maritime pilots are more efficient in generating deep sleep during workweeks, leading to higher amounts of relative deep sleep time (DST%) in workweeks, even though total sleep time (TST) might be limited. We speculate that, in case of increased efficiency, the higher DST% will continue into the first days of the rest week. Second, poor sleep during workweeks could be counteracted by high amounts of rebound sleep. In this case, we expect a higher DST% immediately after the workweek, during the first nights of the rest week. The possible compensatory mechanisms might indicate whether and how maritime pilots are able to recover from periods of poor sleep.

## Materials and methods

### Participants

We used the SCHIP study dataset [21]. The total research population consisted of 19 healthy male maritime pilots (age range: 48 to 60 years), with an average of 18 years of work-related sleep disruption. For the purpose of the ANCHOR study, we used data from 10 maritime pilots. Nine participants had to be excluded for various reasons: development of sleep apnoea (n = 1); retirement (n = 4); technical issues (n = 2), no data available for rest week following a workweek (n = 2) (only rest week preceding the workweek).

Dutch maritime pilots guide international ships into their docking positions in Dutch harbours and work irregular and unpredictable shifts that depend on the amount and variety of arriving ships. Working these shifts mostly results in fragmented sleep divided over multiple sleep sessions per day (24 hours). Sleep disruption in this cohort is defined as a combination of sleep deprivation (missing a full night of sleep due to work), sleep restriction (a shorter night of sleep), and sleep fragmentation (short sleep periods interrupted by calls to work). Detailed information about the maritime pilots and in-/exclusion criteria can be found in the SCHIP methods paper [21]. The SCHIP study was approved by the institutional review board (IRB) (CMO Region Arnhem-Nijmegen, NL55712.091.16; file number 2016–2337) and performed in accordance with good clinical practice (GCP) guidelines and conducted and reported according to the STROBE guidelines for case-control studies. We obtained written informed consent from all participants.

### Sleep measurements

To obtain subjective measurements of sleep characteristics, participants filled out the Pittsburgh Sleep Quality Index (PSQI) with questions regarding bedtimes and wake-up times, sleep latency, total sleep time, sleep efficiency, and sleep disturbances. The PSQI has a maximum score of 21, a total score of 5 was used as cut-off point for sleep disturbances and a score of ≥7 indicates severe/abnormal sleep behaviour [24]. Participants received the instruction to fill out the PSQI twice, once with regard to their estimated scores over the workweeks in the past month and once with regard to their estimated scores over their rest weeks during the past month.

**Home-EEG measurements.** To obtain objective sleep measurements, participants were instructed to wear a dry electrode, single-lead (FpZ-M2) home-EEG device (SmartSleep; Philips, Eindhoven, The Netherlands) for fourteen consecutive days (7 workdays and 7 days off) [25, 26]. In total, ten participants wore the home-EEG device. Seven participants wore the device during two periods and one participant during three periods of a workweek followed by a rest week. Work-related fragmented sleep resulted in multiple sleep sessions per 24 hours. The home-EEG device was originally developed to acoustically stimulate slow-wave sleep,

through automatic EEG-based detection of slow waves in the delta frequency band (0.5–4 Hz). We used the device for measurement purposes only, without auditory stimulation. The (proprietary) SmartSleep algorithm, based on six-second epochs, differentiates between wakefulness, light sleep, and deep sleep [26, 27]. Deep sleep is detected when the root-mean-square of the delta frequency band exceeds a certain threshold and when at least six slow waves are detected in a 20-second window [27]. Even though these sleep stages are calculated based on a single lead, studies have proven feasibility and validity of sleep staging with the home-EEG device [26–31]. The data is expressed as the following sleep characteristics: total sleep time (TST), deep sleep time (DST), wake after sleep onset (WASO), number of arousals and number of awakenings > 5 minutes. Based on these outcome variables deep sleep time was calculated as percentage of TST (DST%) as the main outcome variable.

## Statistical analysis

**Descriptive sleep data.** The descriptive sleep data were assessed for normal distribution by inspection of histograms and the Shapiro-Wilk test. Normally distributed data are shown as mean ± standard deviation (SD), while not-normally distributed data are shown as median with interquartile range (IQR). A paired samples t-test was performed to compare PSQI scores between workweek and rest week. Home-EEG data was analysed using the Wilcoxon signed rank test to compare number of sleep sessions per day, total sleep time (TST) and deep sleep time (DST) between workweeks and rest periods. Alpha was set at 0.05 and tested two-sided. Descriptive data analyses were conducted with IBM SPSS Statistics for Windows, version 20.0 (IBM Corp., Armonk, NY, USA).

**Multilevel models.** The data had a three-level hierarchical structure, with measurement days nested within a 10 day measurement cycle that combined a rest week directly following a workweek, nested within participants. Exploring the fit of increasingly complex models using deviance statistic [32], Bayesian information criterion (BIC), and Akaike Information Criterion (AIC), we built two multilevel models with DST% as the outcome variable to examine which of our two hypotheses most plausibly fit the empirical data.

For our first model, regarding greater efficiency to generate deep sleep during the workweek, we synchronized time on the second day off after a workweek. We fitted a linear spline model that allows both a shift in level and slope on (before and after) the second day off after a workweek. The model allowed for a linear change in DST% during the workweek and the first day of the rest week and an abrupt shift on the second day off with DST% to stay constant for the remaining rest week (i.e., linear slope constrained to zero, as adding a linear slope did not improve model fit). To evaluate our second model, regarding rebound sleep after a workweek, time was synchronized on the last workday. We again fitted a linear spline model, allowing for both a shift in level and slope in DST% on the last workday. Based on model fit, we iterated towards a model in which DST% was held constant during the workweek, and allowed to linearly change during the rest week. For both models, the intercept was allowed to vary over participants (random intercept for participant) and over measurement cycles within participants (random intercept for measurement cycle nested in participant).

No covariates were added to the models, as all participants are male and of similar age and education. Multilevel model analyses were performed in R version 3.6.2 [33].

## Results

We used data from 10 maritime pilots. All participants had the same, high level of education, were Dutch, male and of white European descent (Table 1).

**Table 1. Baseline characteristics.**

| Characteristics | |
|---|---|
| n | 10 |
| Age, years | 51.6 ±2.4 |
| Time of shift work, years | 18.4 ±3.9 |
| BMI, kg/m$^2$ | 25.8 ±2.2 |
| SBP, mmHg | 141 ±15.9 |
| DBP, mmHg | 89.7 ±11.9 |
| Medication use (yes) | 3 (30) |
| Smoking (yes) | 2 (20) |
| History of hypertension | 0 (0) |
| History of high cholesterol | 1 (10) |
| History of diabetes | 0 (0) |

Data are shown as mean ± SD or Number (%).

Abbreviations: BMI, body mass index; SBP, systolic blood pressure; DBP, diastolic blood pressure.

### Descriptive sleep data

Maritime pilots (n = 10) report a mean PSQI score of 3.9 (±2.0) for rest weeks and an average score of 8.2 (±2.2) for workweeks, which was more than twice the score for rest weeks, with values exceeding the validated cut-off point (≥7) for abnormal sleep behaviour (Table 2). The difference resulted from multiple subcomponents of the PSQI (Table 2). Home-EEG recordings calculated per sleep session showed less TST and DST during a workweek compared to a rest week (Table 2). However, when combining the sleep sessions per day, maritime pilots reached a similar amount of TST and slightly less DST in a larger number of sleep sessions during a workweek, compared to a rest week (Table 2). As indicator of improved efficiency to generate deep sleep, the point estimate for DST% was 3.5% higher during the workweek and this estimate was close to statistical significance (p = 0.08).

### Multilevel model analysis (DST%)

Our first model assessed whether maritime pilots are more efficient in generating deep sleep, shown by an increase in DST% during the workweek (Fig 1). As shown in Table 3, during the workweek until the second day of the rest week, DST% increased by 0.6% per day (p = 0.08), peaking at 17.9% at the second day of the rest week. In the remaining rest week, the DST% was constant at a level of 1.5% lower than the peak DST% at the second day of the rest week, though this difference was not statistically significant (p = 0.29). Our second model assessed whether maritime pilots experienced rebound sleep, with an increase in DST% after their workweek (Fig 2). In the resulting model, both the lower DST% during the workweek, and the time-varying DST% during the rest week did not differ, as illustrated in Table 4. In addition, the model fit statistics (AIC/BIC) for model 1 were lower than for model 2.

### Discussion

We examined sleep architecture of maritime pilots in their natural environment using home-based EEG measurements during their workweek and rest week. We explored two hypotheses, one: maritime pilots compensate for poor sleep with increased efficiency in generating deep sleep during workweeks; and two: maritime pilots compensate work-related sleep disruption with excessive rebound sleep in rest weeks. Our results indicate that increased efficiency of

**Table 2. Sleep characteristics.**

|  | Workweek | Rest week | *P*-value |
|---|---|---|---|
| n | 10 | 10 | |
| **PSQI** | | | |
| PSQI, total score | 8.2 ±2.2 | 3.9 ±2.0 | <0.001 |
| PSQI, subjective sleep quality | 1.4 ±0.5 | 0.7 ±0.5 | 0.001 |
| PSQI, sleep latency | 1.7 ±0.7 | 1.0 ±0.7 | 0.03 |
| PSQI, sleep duration | 1.5 ±0.7 | 0.3 ±0.7 | 0.001 |
| PSQI, sleep efficiency | 0.9 ±0.3 | 0.1 ±0.3 | <0.001 |
| PSQI, sleep disturbances | 1.2 ±0.4 | 1.2 ±0.4 | 1.00 |
| PSQI, sleep medication | 0.2 ±0.6 | 0.1 ±0.3 | 0.34 |
| PSQI, daily dysfunction | 1.3 ±0.8 | 0.6 ±0.5 | 0.03 |
| **Home-EEG measurements** | | | |
| Number of sleep sessions per day | 1.3 (1.1–1.8) | 1.0 (1.0–1.0) | 0.03 |
| WASO, min | 30.2 (21.3–42.8) | 30.4 (24.9–52.9) | 0.65 |
| Number of arousals | 29.1 (21.3–30.8) | 34.1 (29.1–37.9) | 0.005 |
| Number of awakenings ≥ 5 minutes | 1.2 (0.8–2.4) | 1.2 (0.5–3.0) | 0.80 |
| Average TST per sleep session, min | 295.0 (221.5–359.9) | 407.6 (343.0–424.8) | 0.005 |
| Average DST per sleep session, min | 38.1 (31.1–61.5) | 53.55 (49.9–68.3) | 0.013 |
| Average DST% per session | 16.3 (12.8–18.5) | 15.6 (12.3–18.9) | 0.96 |
| Average TST per day, min | 409.1 (369.3–432.3) | 419.2 (370.0–428.3) | 1 |
| Average DST per day, min | 58.3 (50.5–70.3) | 65.9 (51.1–73.6) | 0.19 |
| Average DST% per day | 21.9 (20.2–23.6) | 18.4 (13.5–21.4) | 0.08 |
| Average DST% per day, time synchronized on second day of rest week | 20.6 (19.0–23.73) | 17.4 (12.5–22.1) | 0.13 |

Data are shown as mean ±SD or median (IQR).

Abbreviations: PSQI, Pittsburgh Sleep Quality Index (≥5 indicates sleep disturbances, ≥7 indicates severe/abnormal sleep behaviour; WASO, wake after sleep onset; TST, total sleep time; DST, deep sleep time.

generating deep sleep during workweeks is a more likely compensatory mechanism than rebound sleep after workweeks.

In general, maritime pilots report worse sleep quality during workweeks (PSQI) compared to rest weeks, which was confirmed by home-EEG data, showing significantly less TST and absolute DST per sleep session (Table 2). However, multiple sleep sessions are observed in a typical workday. These combined sleep sessions add up to TST and DST slightly lower compared to a day off. This indicates that maritime pilots, while they subjectively experience poor sleep quality, still reach a comparable TST and DST in fragmented sleep sessions over the course of a workday.

The model describing hypothesis 1 represented a better fit with the data and showed a trend towards deeper and thus improved sleep quality. Looking at DST%, we observed a trend towards an increase of 0.6% per day during the workweek, starting with a DST% of 13.8%, rising up to 17.9% in the beginning of a rest week. Even though a 0.6% increase per day does not seem very high, it thereby slowly reaches normal DST% (17.9%). Combined with, on average, a significantly higher DST% in workweeks, we suggest that our data lend more support to hypothesis 1.

In this group of maritime pilots, the compensatory mechanism to counteract sleep disruption of any form (deprivation, fragmentation, restriction) may lie in the ability to become more efficient in generating deep sleep during a workweek. This could explain earlier findings in this cohort [22, 23] of absence of AD-related cognitive decline or amyloid-β accumulation which have been proposed to be linked to poor sleep [8, 12, 13, 15].

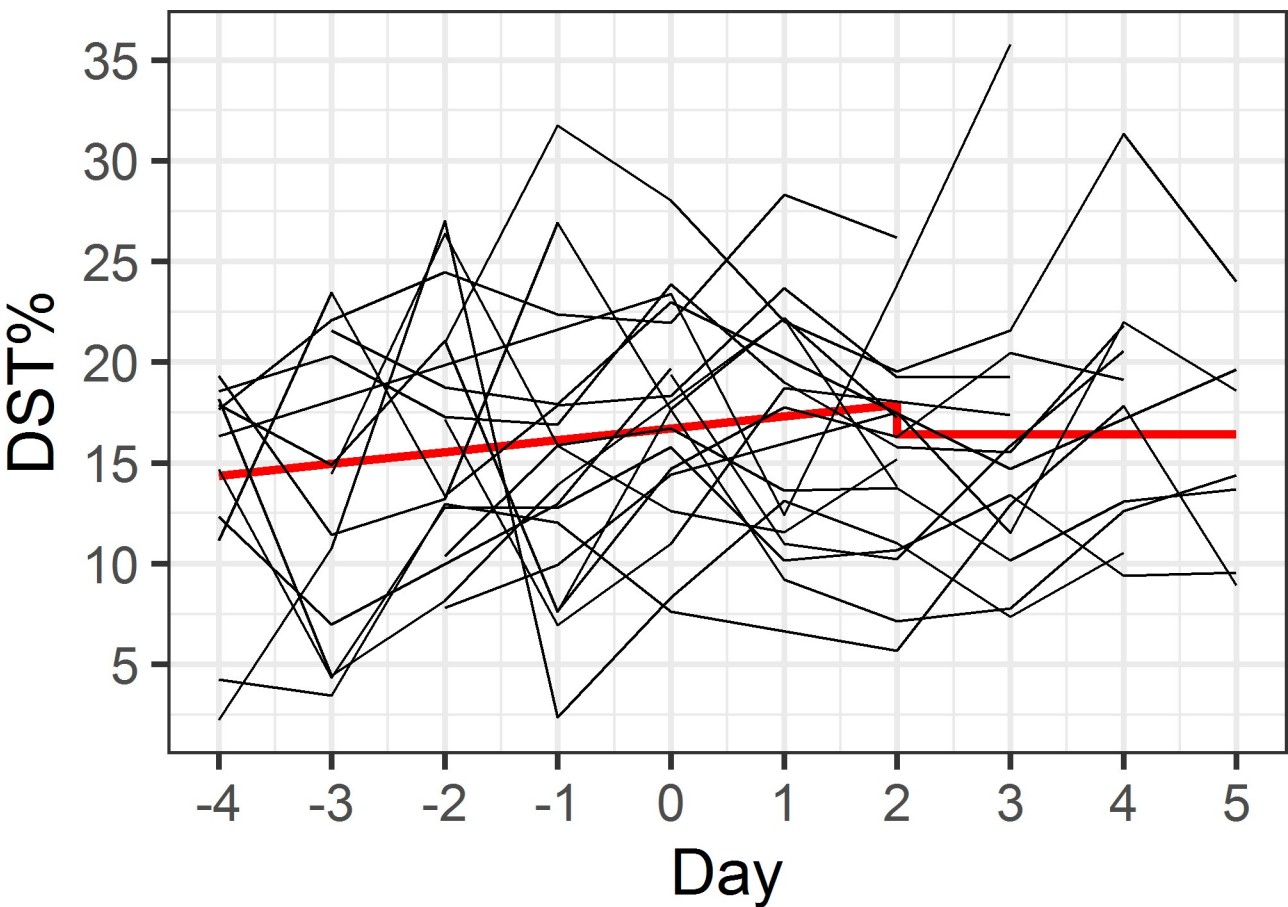

**Fig 1. Model 1 –efficiency in generating deep sleep.** The x-axis indicates consecutive days, with days -4 to 0 representing the workweek and days 1 to 5 representing the rest week. Red line illustrates predicted model values, individual participant data is shown in black.

Borbély and colleagues proposed that deep sleep specifically is enhanced after sleep deprivation [34], which could explain our findings. Ferrara and colleagues showed that relative deep sleep is increased after deep sleep disruption, without any increase in total sleep time, hypothesizing that a fixed amount of deep sleep per night is required rather than sleep duration alone [35]. However, Borbély and Ferrera assessed sleep after total sleep deprivation and selective deep sleep disruption specifically, while we examined sleep architecture during longer periods of sleep disruption. By applying a home-EEG device, which has not been implemented in previous sleep studies, our findings offer more insights into compensatory mechanisms while sleep is disturbed, instead of after sleep disruption has taken place. Thus, we were able to measure sleep architecture during sleep disruption, which has not been feasible in previous studies

**Table 3. Model 1: Increased efficiency to generate deep sleep.**

| Model fit | | AIC | BIC |
|---|---|---|---|
| | | 846.9 | 864.3 |
| Fixed effects | | B (SE) | p-value |
| Average DST% at day 2 of rest week | | 17.9 (1.7) | <0.001 |
| Linear increase in DST% during workweek–day 2 of rest week | | 0.6 (0.3) | 0.08 |
| Difference in DST% during remaining rest week | | -1.5 (1.4) | 0.29 |

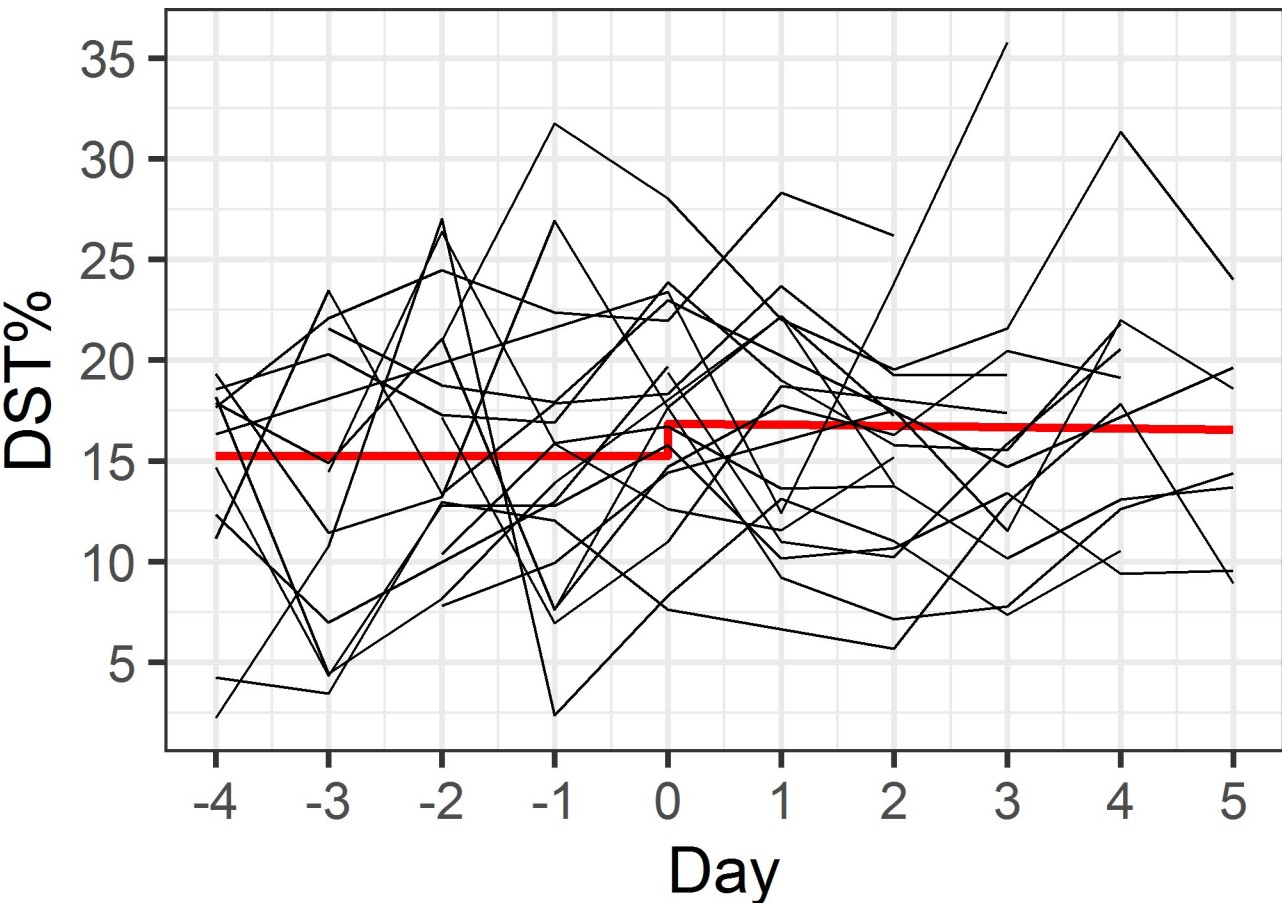

**Fig 2. Model 2 –rebound sleep after workweek.** The x-axis indicates consecutive days, with days -4 to 0 representing the workweek and days 1 to 5 representing the rest week. Red line illustrates predicted model values, individual participant data is shown in black.

due to the nature of sleep assessment. The difference in methodology for sleep assessment, therefore, makes it challenging to compare outcomes of our studies to these of Borbély and Ferrera. Nevertheless, our findings can further be related to sleep actigraphy outcomes from Korsiak and colleagues. They discovered that the daily (24 hours) TST during shift-work was similar to the TST during free time, due to more napping during shift-work, as we have also observed in the maritime pilot cohort (i.e. fragmented sleep sessions over a 24h-period). However, they concluded that shift workers tend to compensate for sleep loss with rebound sleep

**Table 4. Model 2: Rebound sleep after workweek.**

| Model fit | AIC | BIC |
|---|---|---|
| | 847.4 | 864.8 |
| Fixed effects | B (SE) | p-value |
| DST% at switch between work- and rest week | 16.9 (1.5) | <0.001 |
| Constant DST% during workweek | -1.6 (1.1) | 0.16 |
| Linear increase in DST% during rest week | -0.1 (0.4) | 0.87 |

Abbreviations: AIC, Akaike Information Criterion; BIC, Bayesian Information Criterion; DST%, relative deep sleep time.

during free time [36]. This effect was not confirmed in our study: we found no evidence of rebound sleep.

## Strengths & limitations

The ANCHOR study is one of the first studies to examine home-EEG based sleep data, recorded in a home setting for a longer period of time. The use of wearables in a home-setting, instead of polysomnography (PSG) in a sleep laboratory, allowed us to gain insights in sleep patterns during normal workweeks and rest weeks, which otherwise could not have been measured. Combined with additional subjective measures of sleep quality, we were able to comprehensively measure sleep to illustrate the sleep architecture of maritime pilots. The maritime pilot group is a very unique population, as they seem to be more resilient to sleep disruption, evidenced by the fact that they successfully performed their job for approximately 18 years. Their overall (cognitive) health and externally induced sleep disruption allowed us to investigate whether their sleep architecture may be fundamental to this resilience.

The study is limited by the small sample size (n = 10), which has impacted the statistical power of our results. However, the sample size was fixed, as this report pertains to a secondary analysis of the SCHIP study, where we were limited to the maritime pilot group in the Netherlands, who agreed to participate in that extensive study [21]. However, since part of the participants performed multiple measurement cycles of a rest week following a workweek, we were able to include 19 measurement cycles in our analysis. Although we consider the home-EEG measurements a strength of our study, a trade-off was made between a wearable EEG-device that collects limited data and a full PSG, which requires a laboratory environment. The wearable device is a single-lead EEG measurement device with an automated algorithm to calculate sleep staging. Raw data is deleted after each session due to limited storage space and daily retrieval of raw data is not feasible for logistical reasons. Therefore, data is limited compared to PSG, but allows to study sleep architecture over time in participants' natural environment.

## Implications

We discovered some implications for the use of subjective versus objective sleep measurements. While the maritime pilots complained about worse sleep quality (self-reported in PSQI), objective measurements of sleep did not fully confirm this. The discrepancy between objectively and subjectively measured sleep is a well-known issue [37]. Our findings imply that sleep fragmentation is highly relevant for the overall subjective impression of sleep quality. However, detrimental health effects seem unlikely if normal TST and DST can be obtained in multiple sessions, assuming a sufficient level of general health. Still, future studies need to confirm our results and test whether they are generalizable to a larger population. With wearable devices, such as the home-EEG device, large-scale studies in home-settings are now possible [38] to investigate compensatory mechanisms and consequences of poor sleep for the development of neurodegenerative disease and health outcomes in general. For future research, we would therefore recommend to set up longitudinal studies, with inclusion of larger populations of shift workers, as our hypothesis for possible compensatory mechanisms is of importance for a broad population.

## Conclusion

Maritime pilots seem to be more efficient in generating deep sleep when it is most required and might start compensating for sleep loss during the workweek itself, where sleep is still fragmented. The specific intensity and intermittent pattern of sleep disruption in combination with coping mechanisms of the maritime pilot cohort might be protective against detrimental

effects of sleep disruption, such as AD related accumulation of amyloid-β and/or cognitive dysfunction. Results of this study need to be confirmed in future longitudinal studies with comprehensive home-EEG sleep measurements including larger samples and different populations of shift workers.

## Acknowledgments

We would like to thank all participants for taking part in this study and the secretary of the Dutch Maritime Pilot Association for helping with recruitment of participants. Further, we would like to thank Dr. T. Tsoneva and S. Pastoor for their help with data storage and raw data-extraction from the home-EEG devices.

## Author Contributions

**Conceptualization:** Lara J. Mentink, Jana Thomas, Marcel G. M. Olde Rikkert.

**Data curation:** Lara J. Mentink, Jana Thomas.

**Formal analysis:** Lara J. Mentink, Jana Thomas, René J. F. Melis.

**Funding acquisition:** Jurgen A. H. R. Claassen.

**Methodology:** Lara J. Mentink, Jana Thomas, René J. F. Melis.

**Supervision:** Jurgen A. H. R. Claassen.

**Writing – original draft:** Lara J. Mentink, Jana Thomas.

**Writing – review & editing:** Lara J. Mentink, Jana Thomas, René J. F. Melis, Marcel G. M. Olde Rikkert, Sebastiaan Overeem, Jurgen A. H. R. Claassen.

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
