## [Decision Letter · Decision Letter 0]

15 Sep 2020

PONE-D-20-23329

Home-EEG assessment of possible compensatory mechanisms for sleep disruption in highly irregular shift workers – The ANCHOR study

PLOS ONE

Dear Dr. Mentink,

Thank you for submitting your manuscript to PLOS ONE. After careful consideration, we feel that it has merit but does not fully meet PLOS ONE’s publication criteria as it currently stands. Therefore, we invite you to submit a revised version of the manuscript that addresses the points raised during the review process.

- Provide more details on sleep architecture (SWA, delta power etc.).

- Figure legends and methods need more details.

- Provide a power analysis calculation to justify the small sample size.

We look forward to receiving your revised manuscript.

Kind regards,

Henrik Oster, Ph.D.

Academic Editor

PLOS ONE

Additional Editor Comments:

n/a

Journal Requirements:

"This study was funded in parts by the ISAO grant (Internationale Stichting Alzheimer Onderzoek, grant number: 15040). Philips kindly provided the home-EEG devices that were used in this study. The funders had no role in study design, data collection and analysis, decision to publish, or preparation of the manuscript."

4. Thank you for stating the following in the Financial Disclosure section:

"This study was funded in parts by the ISAO grant (Internationale Stichting Alzheimer Onderzoek, grant number: 15040). Philips kindly provided the home-EEG devices that were used in this study. The funders had no role in study design, data collection and analysis, decision to publish, or preparation of the manuscript."

We note that you received funding from a commercial source: Philips

Reviewers' comments:

Reviewer's Responses to Questions

**Comments to the Author**

1. Is the manuscript technically sound, and do the data support the conclusions?

Reviewer #1: Partly

Reviewer #2: Partly

2. Has the statistical analysis been performed appropriately and rigorously? 

Reviewer #1: Yes

Reviewer #2: Yes

3. Have the authors made all data underlying the findings in their manuscript fully available?

Reviewer #1: Yes

Reviewer #2: Yes

4. Is the manuscript presented in an intelligible fashion and written in standard English?

Reviewer #1: Yes

Reviewer #2: Yes

5. Review Comments to the Author

Reviewer #1: Alzheimer’s disease is characterized by an accumulation of amyloid-ß in the brain. It is supposed that clearance of amyloid-ß by the glymphatic system occurs during sleep. It has been shown that disruption of deep sleep led to an increase in amyloid-ß concentration in cerebrospinal fluid. Sleep disruption has been associated with an increased risk for developing Alzheimer’s disease. Earlier studies showed that long-time shift-working pilots with sleep disruption did not show evidence of early Alzheimer’s disease. To explain this finding, the authors hypothesized two mechanisms (1) pilots had increased efficiency in generating deep sleep (DST) during workweek (model 1) and (2) pilots experienced high rebound sleep during rest week (model 2). The authors analyzed data from the SCHIP study dataset from 10 pilots (age 51.6 +/- 2.4 years) with an employment time of 18.4 +/- 3.9 years. The authors applied the Pittsburgh Sleep Quality Index (PSQI) to determine subjective sleep quality of pilots in both workweeks and rest weeks. Objective sleep characteristics were measured by home-EEG devices used by the pilots on 10 workdays and 10 rest days. Results showed that pilots had poorer subjective sleep quality in workweeks than in rest weeks (8.2 vs. 3.9 scores, p<0.001). Average DST per day was shorter in workweeks than in rest weeks (58.3 min vs. 65.9 min, p = 0.19). Average DST% per day was longer in workweeks compared to rest weeks (21.9 vs. 18.4, p = 0.08). Multilevel model analysis, testing model 1 and model 2, indicates that increased efficiency in generating deep sleep during workweeks seems more likely a compensatory mechanism than rebound sleep in rest weeks.

This is an interesting study, but there are some concerns to be addressed.

The main concern refers to the small size of participants (N = 10). Thus, the large difference in “DST per day” between workweeks (58.3 min) and rest weeks (65.9 min) was not statistically significant. Effect size and statistical power should be given.

Shiftwork: It would be helpful if the authors could give more information on the shift work times of the 10 pilots for the 10 days of EEG data collection. Also, it would be interesting to know whether the shiftwork of the 10 pilots was primarily characterized by sleep deprivation or by sleep disruption. This might be of interest for the discussion of the findings.

Measurement of sleep quality:

Page 5, lines 116-117: The authors wrote that the PSQI was completed on a workweek and on a rest week. It would be fine to inform whether the PSQI was completed at the end of the workweek and the rest week. Normally, the PSQI refers to a time period of four weeks.

Home-EEG measurements:

- Page 6, lines 122-123: The authors wrote that some pilots wore the EEG-device during two periods of work days and rest days. Please clarify the number of these pilots. Did the authors compare the results of both periods?

- Table 1: “Employment time” means years of shift work? Please clarify.

- Page 8, lines 175-177: The authors found that pilots had poorer subjective sleep quality (PSQI score) in workweeks than in rest weeks. It would be of interest to know whether the difference in PSQI score was due to differences in sleep onset latency and/or sleep duration between workweeks and rest weeks. Table 2 shows the PSQI score, but does not include sleep duration and sleep-onset latency. The TST in Table 2 refers to EEG measurements, if I correctly understand.

Reviewer #2: The current manuscripts reports findings from a study that sought to test whether long-time shift workers (maritime pilots) compensated for shift related sleep loss by increasing the efficiency of deep sleep during work week sleep sessions or by rebound sleep during the rest week. The results suggest that the pilots have a small increase (that only reached an alpha level of 0.08) in the percent of deep sleep totals (%DST) during the workweek compared to the rest week. However, the pilots acquired significantly less total sleep time and deep sleep time during the work week compared to the rest week. When the data are compared via two computational models to test which may represent a compensatory mechanism for sleep loss (the work vs rest week sleep), the authors concluded that increased efficiency in deep sleep during the work week is more likely represents the compensatory mechanisms.

While the findings are interesting in that they support the classic model that the amount of sleep/sleep efficiency is directly proportional to the time spent awake in prior to sleep, there are some concerns with the study and data presentation.

1. The authors use a novel home EEG monitoring system. It is unclear how well this system can detect Slow Wave Activity (a measure of deep sleep efficiency i.e. delta power) compared to traditional polysomnographic recordings. It would be very useful to the reader if they provided more detailed information concerning delta power and/or a spectral analysis of sleep under the two conditions.

2. The figure legends would benefit from more detail. What are the -days the workweek? are these consecutive? how does the model handle missing days? it appears that several participants do not have a values after day 2, 3, or 4.

3. While the authors acknowledge that the sample size is small (n=10). There is no mention of a power analysis calculation that would provide the reader information about the sample size actually required to reach a power of at least .8

6. PLOS authors have the option to publish the peer review history of their article (what does this mean?). If published, this will include your full peer review and any attached files.

Reviewer #1: No

Reviewer #2: No

---

## [Author Response · Author response to Decision Letter 0]

14 Oct 2020

We thank the reviewers for their time to review our manuscript and for their useful suggestions. Please find our point-by-point response to the reviewer comments below.

Reviewer #1: 

Alzheimer’s disease is characterized by an accumulation of amyloid-ß in the brain. It is supposed that clearance of amyloid-ß by the glymphatic system occurs during sleep. It has been shown that disruption of deep sleep led to an increase in amyloid-ß concentration in cerebrospinal fluid. Sleep disruption has been associated with an increased risk for developing Alzheimer’s disease. Earlier studies showed that long-time shift-working pilots with sleep disruption did not show evidence of early Alzheimer’s disease. To explain this finding, the authors hypothesized two mechanisms (1) pilots had increased efficiency in generating deep sleep (DST) during workweek (model 1) and (2) pilots experienced high rebound sleep during rest week (model 2). The authors analysed data from the SCHIP study dataset from 10 pilots (age 51.6 +/- 2.4 years) with an employment time of 18.4 +/- 3.9 years. The authors applied the Pittsburgh Sleep Quality Index (PSQI) to determine subjective sleep quality of pilots in both workweeks and rest weeks. Objective sleep characteristics were measured by home-EEG devices used by the pilots on 10 workdays and 10 rest days. Results showed that pilots had poorer subjective sleep quality in workweeks than in rest weeks (8.2 vs. 3.9 scores, p<0.001). Average DST per day was shorter in workweeks than in rest weeks (58.3 min vs. 65.9 min, p = 0.19). Average DST% per day was longer in workweeks compared to rest weeks (21.9 vs. 18.4, p = 0.08). Multilevel model analysis, testing model 1 and model 2, indicates that increased efficiency in generating deep sleep during workweeks seems more likely a compensatory mechanism than rebound sleep in rest weeks.

This is an interesting study, but there are some concerns to be addressed.

Major comments:

1. The main concern refers to the small size of participants (N = 10). Thus, the large difference in “DST per day” between workweeks (58.3 min) and rest weeks (65.9 min) was not statistically significant. Effect size and statistical power should be given.

Response: 

We would like to thank the reviewer for the positive remarks. 

For the SCHIP study, we included a unique cohort of maritime pilots with prolonged and consistent sleep disruption due to their work, making this a highly valuable population that allowed us to explore sleep as an isolated variable in relation to Alzheimer’s Disease. This research is a secondary analysis of the SCHIP study, where we were limited to the maritime pilot group in the Netherlands who agreed to participate in that extensive study. As a consequence, the sample size for the current study was fixed. We agree that the present study is limited by the small sample size and that the true power of the study – would we have been able to calculate it in advance for this specific purpose – would have been likely low. However, post-hoc observed power is not informative about the true power and post-hoc power calculations should therefore be refrained from (see http://daniellakens.blogspot.com/2014/12/observed-power-and-what-to-do-if-your.html). As our finding was non-significant, the observed power of the study will also have been low, because “reporting post-hoc power is nothing more than reporting the p-value in a different way” (see blog post from Daniel Lakens: http://daniellakens.blogspot.com/2014/12/observed-power-and-what-to-do-if-your.html).

As for the reporting of the effect size of DST per day, we calculated a point estimate of 0.41 (z-score/√n; 1.31/√10), which qualifies as moderate. However, taking into consideration the standard error of the point estimation, the true effect may be considerably bigger or smaller. Future studies are needed, which can use the current effect size and other considerations to design a study that can provide more definitive answers about the hypothesis that our findings suggest. 

In our study, we were cautious in presenting our results, based on the small sample size, throughout the whole manuscript. We have now extended our limitation section regarding the small sample size on page 14, lines 276-279, or see section below:

The study is limited by the small sample size (n=10), which has impacted the statistical power of our results. However, the sample size was fixed, as this report pertains to a secondary analysis of the SCHIP study, where we were limited to the maritime pilot group in the Netherlands who agreed to participate in that extensive study [21]. 

2. Shiftwork: It would be helpful if the authors could give more information on the shift work times of the 10 pilots for the 10 days of EEG data collection. Also, it would be interesting to know whether the shiftwork of the 10 pilots was primarily characterized by sleep deprivation or by sleep disruption. This might be of interest for the discussion of the findings.

Response: 

We agree it would be interesting to have that information. Unfortunately, we do not have specific information about the individual shift work times. The shifts of maritime pilots are highly dependent on the amount and kind of ships that arrive. During the workweek, maritime pilots work on a rotating schedule, i.e., once the maritime pilots are called to work and finished a trip/shift, they end up on the bottom of the list. They will be called to duty again determined by the rotating schedule, however, it is also dependent on the timing and the kind of ships that arrive. Therefore, the shifts are very random and exact shift work times cannot be predicted or received afterwards. However, all pilots follow this work scheme, thus shift work hours are consistent within the group. 

The shift work of the maritime pilots is furthermore characterized by a combination of sleep deprivation (missing a full night of sleep due to work), sleep restriction (a shorter night of sleep), and sleep fragmentation (short sleep periods interrupted by calls to work). We agreed to use the umbrella term ‘sleep disruption’ throughout the whole manuscript to refer to their unique sleep schedules. The sleep disruption of the maritime pilots therefore, cannot be broken down into specific disturbances like sleep loss or deprivation, but is rather a mixture of different aspects. 

We have now further clarified this in the methods section on page 5, lines 105-109, or see section below.

Working these shifts mostly results in fragmented sleep divided over multiple sleep sessions per day (24 hours). Sleep disruption in this cohort is defined as a combination of sleep deprivation (missing a full night of sleep due to work), sleep restriction (a shorter night of sleep), and sleep fragmentation (short sleep periods interrupted by calls to work). Detailed information about the maritime pilots and in-/exclusion criteria can be found in the SCHIP methods paper [21].

3. Measurement of sleep quality:

Page 5, lines 116-117: The authors wrote that the PSQI was completed on a workweek and on a rest week. It would be fine to inform whether the PSQI was completed at the end of the workweek and the rest week. Normally, the PSQI refers to a time period of four weeks.

Response: 

This is indeed a good point. The PSQI was not completed at a certain time point in a workweek or rest week. Rather, participants received the instruction to fill out the PSQI twice at the same time, once with regard to their estimated scores over the workweeks in the past month and once with regard to their estimated scores over their rest weeks during the past month. 

We adjusted the section about the PSQI in the manuscript on page 6, lines 119-121, or see section below:

Participants received the instruction to fill out the PSQI twice, once with regard to their estimated scores over the workweeks in the past month and once with regard to their estimated scores over their rest weeks during the past month.

4. Home-EEG measurements:

Page 6, lines 122-123: The authors wrote that some pilots wore the EEG-device during two periods of work days and rest days. Please clarify the number of these pilots. Did the authors compare the results of both periods?

Response: 

In total, 10 participants wore the EEG-device, of whom 7 participants wore the device for two periods and 1 participant for three periods of a workweek following a rest week. We did not compare the results of these periods within a participant, because this comparison was not informative to our research question. Our multilevel models however took this fact of multiple measurement periods per participant into account through the implementation of a three- rather than a two level hierarchical model. 

We clarified the number of pilots who wore the EEG-device during multiple periods of a workweek following a rest week in the manuscript on page 6, lines 126-128, or see below:

In total, ten participants wore the home-EEG device. Seven participants wore the device during two periods and one participant during three periods of a workweek following a rest week.

5. Table 1: “Employment time” means years of shift work? Please clarify.

Response: 

Yes, that is correct, the employment time means the years of shift work. We have clarified this in Table 1.

6. Page 8, lines 175-177: The authors found that pilots had poorer subjective sleep quality (PSQI score) in workweeks than in rest weeks. It would be of interest to know whether the difference in PSQI score was due to differences in sleep onset latency and/or sleep duration between workweeks and rest weeks. Table 2 shows the PSQI score, but does not include sleep duration and sleep-onset latency. The TST in Table 2 refers to EEG measurements, if I correctly understand.

Response: 

Indeed, the TST in table 2 refers to the home-EEG measurements, we altered Table 2 to clarify this. Concerning the PSQI subcomponents, this is a very interesting remark. We looked at the subcomponents of the PSQI and found that the difference between the PSQI scores concerning workweeks and rest weeks was due to most factors and not just the difference in sleep onset latency and/or sleep duration. We saw significant differences concerning sleep quality (p=0.001), sleep onset latency (p=0.025), sleep duration (p=0.001), sleep efficiency (p<0.001) and daily dysfunction (p=0.025). 

We added the subcomponents of the PSQI to Table 2 (page 10) and adjusted the section about the PSQI in the manuscript on page 9, lines 185-186, or see section below:

 The difference resulted from multiple subcomponents of the PSQI (Table 2).

Reviewer #2:

The current manuscripts reports findings from a study that sought to test whether long-time shift workers (maritime pilots) compensated for shift related sleep loss by increasing the efficiency of deep sleep during work week sleep sessions or by rebound sleep during the rest week. The results suggest that the pilots have a small increase (that only reached an alpha level of 0.08) in the percent of deep sleep totals (%DST) during the workweek compared to the rest week. However, the pilots acquired significantly less total sleep time and deep sleep time during the work week compared to the rest week. When the data are compared via two computational models to test which may represent a compensatory mechanism for sleep loss (the work vs rest week sleep), the authors concluded that increased efficiency in deep sleep during the work week is more likely represents the compensatory mechanisms.

While the findings are interesting in that they support the classic model that the amount of sleep/sleep efficiency is directly proportional to the time spent awake in prior to sleep, there are some concerns with the study and data presentation.

Major comments:

1. The authors use a novel home EEG monitoring system. It is unclear how well this system can detect Slow Wave Activity (a measure of deep sleep efficiency i.e. delta power) compared to traditional polysomnographic recordings. It would be very useful to the reader if they provided more detailed information concerning delta power and/or a spectral analysis of sleep under the two conditions.

Response:

This is a relevant question. The-EEG device uses one-channel (FpZ-M2) to measure the EEG, which is filtered in three frequency bands: alpha (8-12 Hz), beta (15-30 Hz) and delta (0.5-4 Hz). These three frequency bands are used to detect ‘wake’, ‘light sleep’ and ‘deep sleep’. Deep sleep is detected when the root-mean-square of the delta frequency band exceeds a threshold and when at least six slow waves are detected in a 20 second window. The exact value of the RMS threshold is unknown to us, as it concerns a patented algorithm by the developers. This algorithm has been developed and adjusted using manual sleep scoring of the one-channel EEG by expert sleep technicians (see Garcia-Molina et al, 2018, Closed-loop system to enhance slow-wave activity). The validity of this algorithm has been published, and for example validated as a measure of sleep need dissipation (see Garcia-Molina et al., 2015, Automatic characterization of sleep need dissipation dynamics using a single EEG signal).

To inform the readers, we have added the following to the methods section on page 6, lines 130-136, or see below:

The home-EEG device was originally developed to acoustically stimulate slow-wave sleep, through automatic EEG-based detection of slow waves in the delta frequency band (0.5-4 Hz). We used the device for measurement purposes only, without auditory stimulation. The (proprietary) SmartSleep algorithm, based on 6 second epochs, differentiates between wakefulness, light sleep, and deep sleep [26, 27]. Deep sleep is detected when the root-mean-square of the delta frequency band exceeds a certain threshold and when at least six slow waves are detected in a 20 second window [27].

2. The figure legends would benefit from more detail. What are the -days the workweek? are these consecutive? how does the model handle missing days? it appears that several participants do not have a values after day 2, 3, or 4.

Response:

We agree that the figure legends need some clarification. The days are consecutive, with the rest week starting at day 1. The negative days are the workweek, with day 0 as the last workday. There are participants with missing days, however, these are only present at the start or end of their consecutive days of measurements. We expect these days to be missing completely at random or perhaps missing at random, as not all participants started their consecutive measurements at the first day of the workweek. Mixed models handle missing data of both MCAR and MAR type very well, unlike e.g. a repeated measures ANOVA, participants with missing data on the outcome variable are not excluded from analysis.

We clarified the figure legends in the manuscript, page 12, lines 215-220, or see below:

Fig 1. Model 1 – efficiency in generating deep sleep. The x-axis indicates consecutive days, with days -4 to 0 representing the workweek and days 1 to 5 representing the rest week. Red line illustrates predicted model values, individual participant data is shown in black.

Fig 2. Model 2 – rebound sleep after workweek. The x-axis indicates consecutive days, with days -4 to 0 representing the workweek and days 1 to 5 representing the rest week. Red line illustrates predicted model values, individual participant data is shown in black.

3. While the authors acknowledge that the sample size is small (n=10). There is no mention of a power analysis calculation that would provide the reader information about the sample size actually required to reach a power of at least .8

Response:

Thank you for your comment, that resonates with major comment 1 of reviewer 1 and that we have addressed there. Please refer to our answer there.

---

## [Decision Letter · Decision Letter 1]

12 Nov 2020

Home-EEG assessment of possible compensatory mechanisms for sleep disruption in highly irregular shift workers – The ANCHOR study

PONE-D-20-23329R1

Dear Dr. Mentink,

We’re pleased to inform you that your manuscript has been judged scientifically suitable for publication and will be formally accepted for publication once it meets all outstanding technical requirements.

Kind regards,

Henrik Oster, Ph.D.

Academic Editor

PLOS ONE

Additional Editor Comments (optional):

Reviewers' comments:

Reviewer's Responses to Questions

**Comments to the Author**

1. If the authors have adequately addressed your comments raised in a previous round of review and you feel that this manuscript is now acceptable for publication, you may indicate that here to bypass the “Comments to the Author” section, enter your conflict of interest statement in the “Confidential to Editor” section, and submit your "Accept" recommendation.

Reviewer #2: All comments have been addressed

2. Is the manuscript technically sound, and do the data support the conclusions?

Reviewer #2: Yes

3. Has the statistical analysis been performed appropriately and rigorously? 

Reviewer #2: Yes

4. Have the authors made all data underlying the findings in their manuscript fully available?

Reviewer #2: Yes

5. Is the manuscript presented in an intelligible fashion and written in standard English?

Reviewer #2: Yes

6. Review Comments to the Author

Reviewer #2: The authors were highly responsive to the concerns that were raised. All comments were adequately addressed and the manuscript has been greatly improved. There are no further concerns

7. PLOS authors have the option to publish the peer review history of their article (what does this mean?). If published, this will include your full peer review and any attached files.

Reviewer #2: **Yes: **Jessica A Mong

---

## [Editor Report · Acceptance letter]

21 Dec 2020

PONE-D-20-23329R1 

Home-EEG assessment of possible compensatory mechanisms for sleep disruption in highly irregular shift workers – The ANCHOR study 

Dear Dr. Mentink:

I'm pleased to inform you that your manuscript has been deemed suitable for publication in PLOS ONE. Congratulations! Your manuscript is now with our production department. 

Kind regards, 

on behalf of

Prof. Henrik Oster 

Academic Editor

PLOS ONE